# Effectiveness of a peer educator-coordinated preference-based differentiated service delivery model on viral suppression among young people living with HIV in Lesotho: The PEBRA cluster-randomized trial

Mathebe Kopo[1‡], Thabo Ishmael Lejone[1,2,3‡], Nadine Tschumi[2,3], Tracy Renée Glass[3,4], Mpho Kao[1], Jennifer Anne Brown[2,3], Olivia Seiler[5], Josephine Muhairwe[6], Ntoli Moletsane[7], Niklaus Daniel Labhardt[2,3], Alain Amstutz[2,3]*

1 SolidarMed, Partnerships for Health, Maseru, Lesotho, 2 Division of Clinical Epidemiology, Department of Clinical Research, University Hospital Basel, Basel, Switzerland, 3 University of Basel, Basel, Switzerland, 4 Department of Medicine, Swiss Tropical and Public Health Institute, Allschwil, Switzerland, 5 University of Zurich, Faculty of Medicine, Zurich, Switzerland, 6 USAID, Washington DC, United States of America, 7 Sentebale, Maseru, Lesotho

‡ These authors share first authorship on this work.
* alain.amstutz@unibas.ch

**Data Availability Statement:** A key pseudo-anonymized individual participant dataset collected during the study, along with a data dictionary, is available at the data repository Zenodo with open access. DOI: 10.5281/zenodo.7307834 Link:

## Abstract

### Background

Southern and Eastern Africa is home to more than 2.1 million young people aged 15 to 24 years living with HIV. As compared with other age groups, this population group has poorer outcomes along the HIV care cascade. Young people living with HIV and the research team co-created the PEBRA (Peer Educator-Based Refill of ART) care model. In PEBRA, a peer educator (PE) delivered services as per regularly assessed patient preferences for medication pick-up, short message service (SMS) notifications, and psychosocial support. The cluster-randomized trial compared PEBRA model versus standard clinic care (no PE and ART refill done by nurses) in 3 districts in Lesotho.

### Methods and findings

Individuals taking antiretroviral therapy (ART) aged 15 to 24 years at 20 clinics (clusters) were eligible. In the 10 clinics randomized to the intervention arm, participants were offered the PEBRA model, coordinated by a trained PE and supported by an eHealth application (PEBRApp). In the 10 control clusters, participants received standard nurse-coordinated care without any service coordination by a PE. The primary endpoint was 12-month viral suppression below 20 copies/mL. Analyses were intention-to-treat and adjusted for sex.

From November 6, 2019 to February 4, 2020, we enrolled 307 individuals (150 intervention, 157 control; 218 [71%] female, median age 19 years [interquartile range, IQR, 17 to 22]). At 12 months, 99 of 150 (66%) participants in the intervention versus 95 of 157 (61%) participants in the control arm had viral suppression (adjusted odds ratio (OR) 1.27; 95%

https://zenodo.org/record/7307834#.
Y2wSSXbMI2w.

**Funding:** This trial is predominantly funded by the CIPHER grant (2018) from the International AIDS Society, obtained by AA (https://www.iasociety.org/ias-programme/cipher). The Swiss Institute of Tropical and Public Health is the sponsor of the study. AA received his salary through a grant from the MD-PhD program of the Swiss National Science Foundation (Grant 323530_177576; www.snf.ch). The funders had no role in study design, data collection and analysis, decision to publish, or preparation of the manuscript.

**Competing interests:** The authors have declared that no competing interests exist.

**Abbreviations:** ART, antiretroviral therapy; CATS, community adolescent treatment supporters; CI, confidence interval; COVID-19, Coronavirus Disease 2019; DSD, differentiated service delivery; ICC, intracluster correlation coefficient; IQR, interquartile range; LTFU, loss to follow-up; OR, odds ratio; PE, peer educator; PEBRA, Peer Educator-Based Refill of ART; QoL, quality of life; SMS, short message service; VL, viral load.

confidence interval [CI] [0.79 to 2.03]; $p = 0.327$); 4 of 150 (2.7%) versus 1 of 157 (0.6%) had died (adjusted OR 4.12; 95% CI [0.45 to 37.62]; $p = 0.210$); and 12 of 150 (8%) versus 23 of 157 (14.7%) had transferred out (adjusted OR 0.53; 95% CI [0.25 to 1.13]; $p = 0.099$). There were no significant differences between arms in other secondary outcomes. Twenty participants (11 in intervention and 9 in control) were lost to follow-up over the entire study period. The main limitation was that the data collectors in the control clusters were also young peers; however, they used a restricted version of the PEBRApp to collect data and thus were not able to provide the PEBRA model. The trial was prospectively registered on ClinicalTrials.gov (NCT03969030).

## Conclusions

Preference-based peer-coordinated care for young people living with HIV, compared to nurse-based care only, did not lead to conclusive evidence for an effect on viral suppression.

## Trial registration

clinicaltrials.gov, NCT03969030, https://clinicaltrials.gov/ct2/show/NCT03969030.

## Author summary

### Why was this study done?

- An estimated 2.1 million young people aged 15 to 24 years old are living with HIV in Southern and Eastern Africa. Compared to younger children or adults living with HIV, young people have worse viral suppression rates and are at greater risk of dying and being lost from care.

- Although this population has unique needs, preference-based models of care that offer a wide range of service options that are regularly adapted to the young persons' preferences are scarce, and data about their effectiveness are limited.

### What did the researchers do and find?

- The PEBRA (Peer Educator Based Refill of ART) care model was developed during a series of formative workshops in close collaboration with peer educators (PEs), young people living with HIV, youth activists, clinicians, and young mobile application developers, focussing on rural health clinics in Lesotho.

- In the PEBRA model, a PE regularly assessed service preferences of his/her peers regarding medication pick-up, short message service (SMS) notifications, and psychosocial support and delivered services accordingly.

- This cluster-randomized trial compared PEBRA model versus standard clinic care (no PE-coordinated care) in 3 districts in Lesotho.

- At 12 months, 66% and 61% participants in the intervention and control arm, respectively, achieved viral suppression with no significant difference between arms.

- We found that arms were similar in terms of engagement in care, transfer out, self-reported adherence, quality of life (QoL), and satisfaction with care. Four and 1 death occurred in intervention and control arm, respectively.

### What do these findings mean?

- This trial assessing a preference-based peer-coordinated care model for young people living with HIV, compared to nurse-based care only, did not demonstrate conclusive evidence for an effect on viral suppression. Large-scale research is needed to tackle the insufficient viral suppression rate among young people living with HIV in Lesotho.

## Introduction

An estimated 2.1 million young people aged 15 to 24 years old are living with HIV in Southern and Eastern Africa [1]. Young people have poorer outcomes compared to adults at every stage of the HIV care cascade, resulting in virological failure and a high HIV-related mortality [2–4].

Differentiated service delivery (DSD) is a person-centered approach in HIV care that aims to offer services according to the specific needs of key groups of people living with HIV [5]. DSD has been widely implemented for adults living with HIV with promising results on clinical outcomes, but DSD models and data for the younger population groups are scarce [6]. In Lesotho, similar to other countries in the region, efforts to adapt HIV care to adolescents and young adults are promoted [7], but young peers still only play a limited role in providing services.

Several systematic reviews concluded that evidence for interventions to improve engagement in HIV care among adolescents and young adults in low-resource settings is of low quality [6,8,9], and in general, DSD models are not designed for young people nor led by young people [10]. The Zvandiri trial in Zimbabwe demonstrated that adolescents benefit from community adolescent treatment supporters [11]; however, a peer educator (PE)-coordinated service delivery at the facility has not been evaluated yet.

Together with PEs, young people living with HIV, youth activists, clinicians, district Ministry of Health authorities, and mobile application developers, the PEBRA (Peer Educator-Based Refill of ART) model was developed. In PEBRA, a PE at the health facility assisted by a dedicated eHealth application, assessed regularly patients' preferences regarding medication pick-up, short message service (SMS) notifications, and psychosocial support and delivered services accordingly.

The PEBRA cluster-randomized trial compared the PEBRA model versus standard clinic care, where no PE-coordinated care was offered, in 3 districts in Lesotho. We hypothesized that PEBRA will have a beneficial effect on viral suppression and other clinical and HIV care-related outcomes.

## Methods

### Study design and participants

The PEBRA trial was a cluster-randomized, open label, pragmatic clinical trial conducted at 20 rural nurse-led health facilities (clusters) in Butha-Buthe, Leribe, and Mokhotlong districts of

Lesotho, Southern Africa. The 20 health facilities serve a rural population in a mountainous area with poor infrastructure. We decided to conduct this trial only across nurse-led clinics in rural areas because these are the most common health facilities across Lesotho, they provide standardized services (e.g., important for the control arm comparison) and the population they serve reports generally worse HIV outcomes compared to the population served at the district hospitals [12]. Recruitment for the trial lasted from November 6, 2019 until February 4, 2020. A detailed study protocol has been published previously [13].

Eligible clusters were public or missionary nurse-led clinics (not hospitals) offering ART services to a rural population, situated in an area with stable cell phone signal, willing to participate, and with a PE who passed the study-specific training assessment. Young people were eligible for inclusion if they were living with HIV, registered for HIV care at one of the eligible clusters, aged 15 to 24 years, taking ART, and able to provide informed consent.

### Randomization and blinding

Randomization events with all health facilities and the District Health Management Team were conducted in each district between October 25 and 30, 2019. At these events, health facility representatives drew opaque, sealed, equally sized envelopes containing the group allocation (control or intervention) from a Mokorotlo (traditional Lesotho hat), and disclosure took place only once all facilities had drawn their envelope. Additionally, to minimize potential selection bias, the sequence of drawing was randomly selected in advance by an independent person drawing from a second pile of opaque, sealed envelopes containing the names of the facility. Randomization was stratified by district (Butha-Buthe versus Leribe versus Mokhotlong), using 1 block per strata. Participating clinics were assigned (1:1) to either offer standard of care or the PEBRA model. This was an open-label trial; however, laboratory staff who assessed the primary endpoint were blinded. We used non-site-specific study ID numbers on all laboratory and data collection forms to maintain masking.

### Procedures

During the recruitment period, the PE actively screened all young people attending their health facility for inclusion on a rolling basis and obtained written informed consent. The PEs recruited from Monday to Saturday on a strictly rolling basis and followed a screening log file. Recruitment happened at all 20 health facilities concurrently. Illiterate participants provided a thumbprint and chose a literate witness (independent of the trial and chosen by the participant) to co-sign the form. In order to minimize selection bias, the ethics committees agreed to waive parental consent for the 15 to 17 years old study participants. If eligible, the PE administered a questionnaire that included sociodemographic and socioeconomic data, medical history, adherence to ART, quality of life (QoL), HIV/AIDS-related knowledge, and satisfaction with care. If no viral load (VL) within the previous 12 months was available, the participant was sent to the nurse for VL measurement.

### Intervention and control

At health facilities randomized to the intervention arm, the participants were offered the PEBRA model. PEBRA is a DSD model coordinated by the PE, delivered using an eHealth application (PEBRApp) and based on regular service preference assessments in 3 domains: (1) medication pick-up; (2) SMS notifications; and (3) psychosocial support. An overview of the PEBRA model is presented in Fig 1 and details to each intervention component in S1 Table.

First, the PE explained all options within each domain to the participant (Fig 1). Second, the participant picked his/her preferred choice within each domain. Third, the PE

systematically assessed the feasibility of the options chosen, as not all options are available to everyone all the time, e.g., no nearby Village Health Worker available who could dispense ART, or no community youth club established in the participants' community, or home-delivery by the PE not feasible. Finally, the compromise between preference and feasibility was delivered. This process was guided by the PEBRApp, an android-based application, installed on the tablet of the PEs.

The PEBRA model and the PEBRApp were developed in close collaboration with PEs, young people living with HIV, youth activists, clinicians, district Ministry of Health authorities, Sentebale youth leaders, SolidarMed and Swiss Tropical and Public Health Institute research staff as well as local and international mobile application developers. During a series of workshops, these stakeholders defined the current main challenges in adolescent HIV care in Lesotho, possible strategies within the PEBRA model to overcome these challenges building on existing resources at the study health facilities, the role of the PE within the PEBRA model, and the scope and design of the PEBRApp. Due to limited access to smartphones in rural Lesotho, the consensus from the formative workshop series was to build an application for the PEs only, with a communication channel between PE and participants based on SMS technology. As such, the PEBRApp helped the PE not only to collect data, regularly assess the preferences of their peers, and to keep track of the ART refill and next assessment dates, but also ensured regular contact between the PE and the participant. More details and screenshots about the PEBRApp are provided in the published study protocol [13] and the code is open-source available on github (https://github.com/chrisly-bear/PEBRApp). Any selected SMS notifications were sent out automatically according to chosen frequency and content and always included a free-of-charge call-back option to the PE's phone number. These service preferences were assessed at enrolment and thereafter every month for participants with unsuppressed VL ($\geq$1,000 copies/mL) and every 3 months for participants with suppressed VL.

Before starting the trial, the PEs completed a 1-week training camp that covered obtaining informed consent, administering study-specific questionnaires, the use of the PEBRApp, and how to deliver PEBRA model. The PEs were recruited from Sentebale's longstanding PE program in collaboration with the Ministry of Health and the SolidarMed study coordinator, received a monthly stipend and were closely supervised by the attached health facility staff and the study coordinator.

At health facilities randomized to the control arm, the study participants received the standard of care offered at nurse-led rural health facilities in Lesotho, i.e., ART refill at the clinic and nurse-led support (Fig 1).

## Data collection

In intervention arm, data were collected by the PE and in control arm by trained young study staff members. In both arms, data were entered into the password-protected PEBRApp. However, the PEBRApp in control arm was limited to baseline and follow-up data collection and thus did not allow any DSD. The randomization assignment of the health facilities was preloaded into the PEBRApp, and unique individual identifiers were automatically generated. The PEBRApp was connected to the routine VL laboratory platform that includes all VL measurements from the 2 study districts. An automatic synchronization ensured regular download of VL measurements of study participants as well as anonymized data upload to a password-protected, secured database that could be accessed by the data manager. The Satisfaction with Care questionnaire including satisfaction with the PEBRA model was administered by the study staff, all other questionnaires by the PEs. Data integrity checks were programmed into the PEBRApp and the data manager monitored the uploaded data on a regular basis. Data closure was on April 7, 2021.

| | Intervention (PEBRA model) | Control (Standard of care) | Comment |
|---|---|---|---|
| **ART refill** | | | |
| At the clinic | ✓ | ✓ | At regular refill visits or during the Saturday Clinic Club |
| | | ✓ | |
| At the Community Adherence Club | ✓ | ✓ | |
| By a Treatment Buddy | ✓ | ✗ | |
| By the Village Health Worker | ✓ | ✗ | |
| By the Peer Educator | ✓ | | |
| **SMS notifications** | | | |
| Adherence reminders | ✓ | ✗ | Individualized message at preferred time of day |
| ART refill reminders | ✓ | ✗ | Individualized message at preferred time of day |
| Viral load result message | ✓ | ✓ | Individualized message according to viral load result |
| **Psychosocial & informational support options** | | | |
| Nurse at the clinic | ✓ | ✓ | |
| Saturday Clinic Club | ✓ | ✓ | |
| Community Youth Club | ✓ | ✓ | |
| Phone call by Peer Educator | ✓ | ✗ | |
| Home visit by Peer Educator | ✓ | ✗ | |
| School talk by Peer Educator | ✓ | ✗ | |
| Pitso talk by Peer Educator | ✓ | ✗ | Pitso is a village gathering |
| Condom demonstration | ✓ | ✗ | |
| More information about family planning | ✓ | ✗ | |
| More information about voluntary male medical circumcision | ✓ | ✗ | |
| More information about legal aid and gender-based violence | ✓ | ✗ | |
| Linkage to young mothers' group | ✓ | ✗ | |
| Linkage to a female social asset building model | ✓ | ✗ | |

**Fig 1. Description of the PEBRA model and clusters. ART, antiretroviral therapy; PEBRA, Peer Educator-Based Refill of ART.**

## Outcomes

The primary outcome was viral suppression at 12 months, defined as the proportion of all participants in care with a VL below 20 copies/mL at 12 months (range 9 to 15 months) after enrolment. This also included participants who transferred out to another health facility and had a documented VL from the new facility. Blood draw for VL measurement for all study participants was conducted at the clinics using full blood, and the analysis was performed at the corresponding laboratories of the study districts using COBAS TaqMan HIV-1 Test, v2.0 (Roche Diagnostics).

The secondary endpoints were viral suppression using the by then valid WHO threshold of less than 1,000 copies/mL at 12 months, engagement in care at 6 (range 5 to 8) and 12 months, transfer out to another health facility, loss to follow-up (LTFU) (defined as more than 2 months late for a scheduled visit or medication pick-up and no information found about the participant), all-cause mortality, perfect self-reported adherence to ART in the past month (defined as no missed doses according to self-reporting), physical and mental QoL measured using the Short Form 12 (SF-12) questionnaire, satisfaction with care (based on a setting-validated HIV service satisfaction questionnaire [14]), and satisfaction with their PE (in intervention arm only). Details about the endpoints are provided in the published study protocol [13].

## Statistical analysis

According to routine cohort data from the study districts [12], we estimated to recruit 10 to 20 eligible participants per site. For the control arm, based on the same routine cohort data, we

assumed a viral suppression rate of 70% in Butha-Buthe and Leribe districts, but significantly lower in Mokhotlong district. We hypothesized that the PEBRA model would increase the proportion with viral suppression by 20 percentage points in the intervention arm. Using a power of 90%, assuming a type 1 error of 0.05 and a conservative intracluster correlation coefficient (ICC) of 0.05 (design effect of 1.7) based on similar studies [15,16], a sample size of 300 participants in 20 clusters (10 per arm) was needed.

The analysis followed an intention-to-treat approach, including all enrolled participants as randomized. Health facilities (clusters) were the unit of randomization, whereas individuals were the unit of analysis, with viral suppression as a binary outcome. All participants missing their blood draw or having invalid VL results were classified as not meeting the viral suppression endpoints. For the analysis of the primary endpoint, we used a multilevel logistic regression model including cluster (health facility) as a random effect, and arm allocation and the randomization stratification factor (district) as a fixed effect. According to the procedure outlined in our statistical analysis plan, we first assessed all baseline tables for clinically important random imbalanced factors. We noted a substantial random imbalance in sex and thus decided to adjust for it in all analyses.

For the secondary endpoints, we used the complete case set. The secondary endpoints of alternative viral suppression threshold, engagement in care, transfer out, LTFU, and mortality were analysed using a logistic regression model with the same explanatory variables as the primary analysis model. To compare perfect adherence between the 2 arms, we used the same model, but additionally adjusted for self-reported perfect adherence at enrolment and the treatment taken at the time point under consideration (dolutegravir (DTG) versus non-dolutegravir). Differences of QoL between the 2 arms were assessed using multilevel mixed effect linear regression models, adjusted for sex and additionally the respective QoL scores at enrolment. Similarly, satisfaction with care was assessed using a multilevel mixed effect logistic regression model with outcome "very satisfied with care" at 12 months, adjusted for sex and additionally the respective satisfaction at baseline. From the logistic regression models, results are presented as odds ratios (ORs) with 95% confidence interval (CI) and from the linear regression model as beta coefficients with 95% CI. We performed prespecified sensitivity analyses for the primary endpoint: adjustment for baseline VL, using a wider primary endpoint visit window (9 to 18 months), and restricting to the individual per protocol set (participants who attended both the 6-month and 12-month study visit). For the primary outcome, we also assessed effect modification by prespecified variables (age groups, sex, marital status, occupational status, time of ART exposure, DTG at time of endpoint). All analyses were done using R, version 4.0.3 (2020-10-10).

## Ethics statement

This trial was approved by the National Health Research and Ethics Committee of the Ministry of Health of Lesotho (118–2019; June 3, 2019) and the ethics committee in Switzerland (Ethikkommission Nordwest- und Zentralschweiz; 2019–00480; June 14, 2019). The trained study staff obtained the individual written informed consent from the participants before inclusion into the PEBRA trial. To minimize selection bias, the ethics committees agreed to waive parental consent for the 15 to 17 years old study participants as outlined in the approved PEBRA study protocol [13]. Illiterate study participants provided a thumbprint and a literate witness (independent to the trial and chosen by the participant) co-signed the form. The informed consent was provided in the local language, Sesotho, and the participant received a copy of the consent form. Participants were not compensated for participation. The PEBRA trial is registered with ClinicalTrials.gov (https://clinicaltrials.gov/ct2/show/NCT03969030; prospectively

registered on May 31, 2019) and reported as per the CONSORT extension for Cluster Trials guidelines (S1 CONSORT Checklist).

## Results

In October 2019, we identified 25 health facilities in the 3 study districts. Five health facilities were not eligible (Fig 2). The remaining 20 health facilities (clusters) were randomly assigned, 1:1, to deliver either the PEBRA model (intervention arm) or the standard of care (control arm). Three clinics in the control arm and 2 in the intervention arm were missionary facilities. Between November 6, 2019 and February 4, 2020, 315 young people (aged 15 to 24 years) living with HIV and receiving ART care at the 20 clusters were identified and approached. Of these, 8 (2.5%) declined enrolment, 4 in each group. It total, 307 participants—150 in intervention clusters and 157 in control clusters—were enrolled and included in the intention-to-treat analysis (Fig 2).

The baseline characteristics of the 307 participants are shown in Table 1. The median age was 19 years (interquartile range [IQR] 17 to 22), 218 (71%) female, 93 (30%) were married, 107 (35%) had at least 1 child, 22 (7.2%) were (self-)employed with regular income, and 93 (30%) were attending school at time of enrolment. At enrolment, 40 (13%) were taking a DTG-based regimen, the median time taking ART was 3.7 years (IQR 1.7 to 8.4) and 166 (54%) participants had an undetectable baseline VL (<20 copies/mL).

At the 12-month follow-up, 99 of 150 (66%) participants in the intervention versus 95 of 157 (61%) participants in the control arm achieved viral suppression below 20 copies/mL (adjusted OR 1.27; 95% CI [0.79 to 2.03]; $p$ = 0.327; Table 2). Of all 258 participants in care, 22 (8.5%) had a missing VL in the primary endpoint window (11 participants in each arm) and were therefore classified as having an unsuppressed VL (Fig 2).

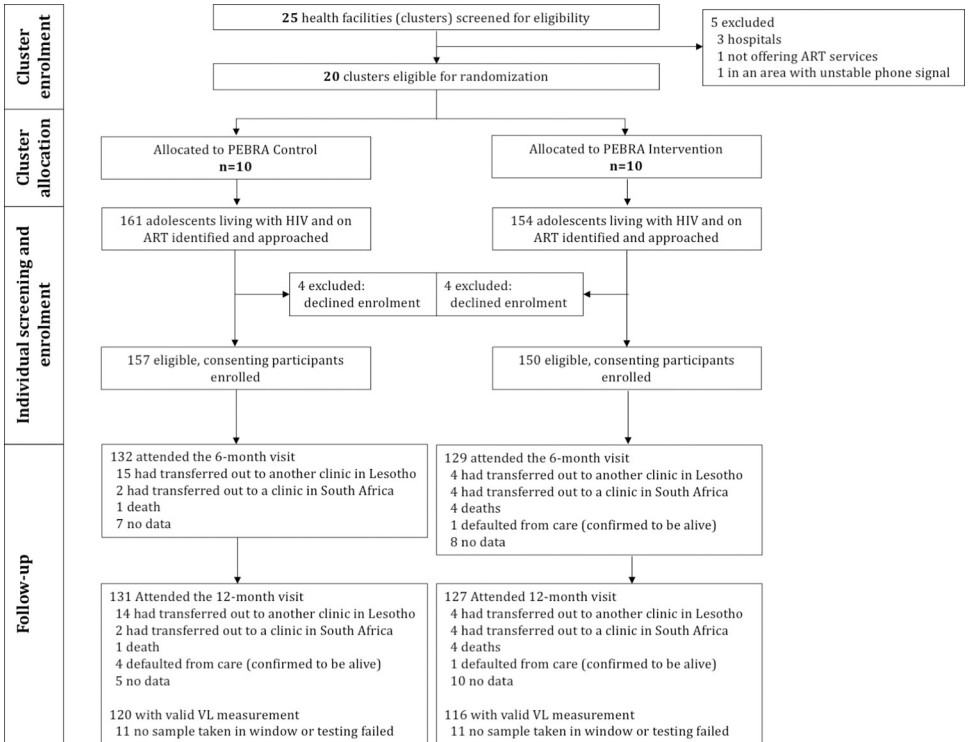

**Fig 2. Consort flow diagram.** ART, antiretroviral therapy; PEBRA, Peer Educator-Based Refill of ART;, VL, viral load.

**Table 1. Baseline characteristics of study participants.**

| | Level | Total | Control | Intervention |
|---|---|---|---|---|
| | | *n* = 307 | *n* = 157 | *n* = 150 |
| Sex | Female | 218 (71.0) | 119 (75.8) | 99 (66.0) |
| Age in years, median (IQR) | | 19.41 [16.94, 22.44] | 20.12 [17.03, 22.94] | 18.72 [16.81, 22.07] |
| Sexual orientation | Heterosexual | 304 (99.0) | 156 (99.4) | 148 (98.7) |
| | Homosexual | 1 (0.3) | 0 (0.0) | 1 (0.7) |
| | Prefer not to answer | 2 (0.7) | 1 (0.6) | 1 (0.7) |
| Has regular access to a cell phone to receive confidential information | Yes | 201 (65.5) | 105 (66.9) | 96 (64.0) |
| Number of completed school years, median (IQR) | | 9.00 [7.00, 10.50] | 9.00 [7.00, 11.00] | 9.00 [7.25, 10.00] |
| Primary occupation | (self-)employed with regular income | 22 (7.2) | 9 (5.7) | 13 (8.7) |
| | Attending school | 93 (30.3) | 36 (22.9) | 57 (38.0) |
| | None of the above | 192 (62.5) | 112 (71.3) | 80 (53.3) |
| Profession (if employed or self-employed) | Business man/woman | 3 (1.0) | 2 (1.3) | 1 (0.7) |
| | Domestic worker | 4 (1.3) | 3 (1.9) | 1 (0.7) |
| | Herdboy | 3 (1.0) | 0 (0.0) | 3 (2.0) |
| | Other* | 12 (3.9) | 4 (2.5) | 8 (5.3) |
| | NA | 285 (92.8) | 148 (94.3) | 137 (91.3) |
| Marital status | Single | 206 (67.1) | 99 (63.1) | 107 (71.3) |
| | Married | 93 (30.3) | 54 (34.4) | 39 (26.0) |
| | Divorced/separated/widowed | 8 (2.6) | 4 (2.5) | 4 (2.7) |
| Pregnant or breastfeeding | No | 184 (59.9) | 104 (66.2) | 80 (53.3) |
| | Yes | 34 (11.1) | 15 (9.6) | 19 (12.7) |
| | NA (male) | 89 (29.0) | 38 (24.2) | 51 (34.0) |
| Number of children | 0 | 200 (65.1) | 91 (58.0) | 109 (72.7) |
| | 1 | 81 (26.4) | 46 (29.3) | 35 (23.3) |
| | 2 | 24 (7.8) | 18 (11.5) | 6 (4.0) |
| | 3 | 2 (0.7) | 2 (1.3) | 0 (0.0) |
| Contraception use | Yes | 134 (43.6) | 85 (54.1) | 49 (32.7) |
| | No | 86 (28.0) | 18 (11.5) | 68 (45.3) |
| | Not currently sexually active | 64 (20.8) | 42 (26.8) | 22 (14.7) |
| | I prefer not to answer | 23 (7.5) | 12 (7.6) | 11 (7.3) |
| What kinds of contraception (if contraception used; multiple choice) | Male or female condom | 91 (29.6) | 62 (39.5) | 29 (19.3) |
| | Contraceptive pill | 18 (5.9) | 11 (7.0) | 7 (4.7) |
| | Injectable or Implant | 42 (13.7) | 24 (15.3) | 18 (12.0) |
| | Withdraw | 4 (1.3) | 3 (1.9) | 1 (0.7) |
| | Calendar method | 3 (1.0) | 3 (1.9) | 0 (0.0) |
| | No answer | 1 (0.3) | 1 (0.6) | 0 (0.0) |
| Years since HIV diagnosis, median (IQR) | | 4.52 [1.86, 9.74] | 3.63 [1.52, 7.87] | 5.45 [2.91, 10.99] |
| Years since starting ART, median (IQR) | | 3.65 [1.65, 8.39] | 3.14 [1.21, 5.82] | 4.90 [2.67, 9.35] |
| Current ART regimen | EFV-based | 212 (69.1) | 104 (66.2) | 108 (72.0) |
| | NVP-based | 47 (15.3) | 22 (14.0) | 25 (16.7) |
| | LPV/r-based | 8 (2.6) | 5 (3.2) | 3 (2.0) |
| | DTG-based | 40 (13.0) | 26 (16.6) | 14 (9.3) |
| Currently receiving treatment for tuberculosis | Yes | 8 (2.6) | 6 (3.8) | 2 (1.3) |
| CD4 count at ART start | <200 | 36 (11.7) | 13 (8.3) | 23 (15.3) |
| | 200–499 | 53 (17.3) | 32 (20.4) | 21 (14.0) |
| | >499 | 39 (12.7) | 24 (15.3) | 15 (10.0) |

(*Continued*)

**Table 1.** (Continued)

| | Level | Total | Control | Intervention |
|---|---|---|---|---|
| | Missing | 179 (58.3) | 88 (56.1) | 91 (60.7) |
| Baseline viral load | <20 | 166 (54.1) | 84 (53.5) | 82 (54.7) |
| | 20–999 | 63 (20.5) | 30 (19.1) | 33 (22.0) |
| | >999 | 44 (14.3) | 26 (16.6) | 18 (12.0) |
| | Missing | 34 (11.1) | 17 (10.8) | 17 (11.3) |
| How do you believe you were infected with HIV? | Blood products | 17 (5.5) | 11 (7.0) | 6 (4.0) |
| | I do not know | 103 (33.6) | 50 (31.8) | 53 (35.3) |
| | I prefer not to answer | 6 (2.0) | 5 (3.2) | 1 (0.7) |
| | Other | 7 (2.3) | 7 (4.5) | 0 (0.0) |
| | Sex with a man | 63 (20.5) | 36 (22.9) | 27 (18.0) |
| | Sex with a woman | 4 (1.3) | 2 (1.3) | 2 (1.3) |
| | Through my mother | 107 (34.9) | 46 (29.3) | 61 (40.7) |
| Has to spend money for transport to the clinic | Yes | 103 (33.6) | 53 (33.8) | 50 (33.3) |
| Transport costs, one way, in Maloti (if transport expenses incurred), median (IQR) | | 17.00 [10.00, 30.00] | 16.00 [8.00, 35.00] | 20.00 [10.00, 30.00] |
| Has to spend money for food during clinic attendance day | Yes | 47 (15.3) | 27 (17.2) | 20 (13.3) |
| Food costs, in Maloti (if food expenses incurred), median (IQR) | | 10.00 [6.00, 20.00] | 10.00 [8.00, 20.00] | 10.00 [5.00, 21.25] |

Results are n (% of those with non-missing data) for categorical variables and median (IQR) for continuous variables.

*Other professions include construction worker, driver, farmer, and unknown.

ART, antiretroviral therapy; DTG, dolutegravir; EFV, efavirenz; IQR, interquartile range; LPV/r, lopinavir/ritonavir; NVP, nevirapine.

The primary analysis indicated that participants from Mokhotlong district were less likely to reach viral suppression. This effect was no longer observed when we adjusted for baseline VL in the prespecified sensitivity analysis (S2 Table). The primary endpoint was consistent across the other 2 prespecified sensitivity analyses (S3 Table).

We found no significant interaction of the prespecified effect modifiers on the primary endpoint (S4 Table). At the end of the study, 268/307 (87%) participants were taking DTG-based ART as compared to 13% at the beginning.

Twelve of 150 (8%) in the intervention versus 23 of 157 (14.7%) in the control arm had transferred out at 12 months (aOR 0.53; 95% CI [0.25 to 1.13]; $p = 0.099$; Table 2). The PEBRA model had a favorable effect on most other secondary endpoints, but the differences were not significant (Table 2). Of note, there were 4 (2.7%) deaths in the intervention versus 1 (0.6%) in the control arm (aOR 4.12; 95% CI [0.45 to 37.62]; $p = 0.210$). None of the deaths were judged to be related to study procedures.

## Discussion

The PEBRA trial was a pragmatic cluster-randomized clinical trial to assess the effectiveness of a youth DSD model, whereby PEs supported young people living with HIV, regularly asked them about their service preferences in terms of SMS notifications, ART refill location, and psychosocial support and delivered care accordingly.

Overall, viral suppression increased from 166/307 (54%) at baseline to 194/307 (63%) at 12 months of follow-up. However, there was no conclusive evidence of a significantly higher rate of viral suppression at 12 months among participants in the intervention clusters compared to standard of care. We found that arms were similar in terms of engagement in care, transfer out, self-reported adherence, QoL, and satisfaction with care.

**Table 2. Primary and secondary endpoints.**

| | Total | Control | Intervention | Adjusted OR or linear regression coefficient; *p*-value (95% CI) [1] | Unadjusted OR or linear regression coefficient; *p*-value (95% CI) [1,2] |
|---|---|---|---|---|---|
| | *N* = 307 | *N* = 157 | *N* = 150 | | |
| *Primary endpoint* | | | | | |
| VL <20 copies/mL [3] | 194 (63%) | 95 (61%) | 99 (66%) | 1.27 (0.79 to 2.03); 0.327 | 1.28 (0.8 to 2.04); 0.304 |
| *Secondary endpoints* | | | | | |
| VL <1,000 copies/mL at 12 mo [3] | 227 (74%) | 114 (73%) | 113 (75%) | 1.14 (0.68 to 1.91); 0.627 | 1.16 (0.7 to 1.95); 0.564 |
| Engagement in care at 6 mo | 261 (85%) | 132 (84%) | 129 (86%) | 1.13 (0.6 to 2.12); 0.714 | 1.17 (0.62 to 2.19); 0.631 |
| Engagement in care at 12 mo | 258 (84%) | 131 (83%) | 127 (85%) | 1.05 (0.56 to 1.94); 0.889 | 1.10 (0.6 to 2.04); 0.751 |
| Transfer out at 12 mo | 35 (11%) | 23 (15%) | 12 (8%) | 0.53 (0.25 to 1.13); 0.099 | 0.50 (0.2 to 2.65); 0.068 |
| LTFU at 12 mo | 20 (7%) | 9 (6%) | 11 (7%) | 1.33 (0.53 to 3.33); 0.538 | 1.30 (0.52 to 3.24); 0.571 |
| All-cause mortality at 12 mo [4] | 5 (2%) | 1 (1%) | 4 (3%) | 4.12 (0.45 to 37.62); 0.210 | 4.27 (0.47 to 38.68); 0.196 |
| Perfect self-reported adherence at 12 mo [5,6] | 123 (40%) | 63 (40%) | 60 (40%) | 1.92 (0.49 to 7.52); 0.348 | 1.69 (0.47 to 6.08); 0.424 |
| Median physical QoL score, median (IQR) [6] | 41.45 (40.21–43.59) | 41.05 | 41.93 | 0.87 (−0.51 to 2.24); 0.235 | 0.89 (−0.49 to 2.27); 0.225 |
| Median mental QoL score, median (IQR) [6] | 46.99 (44.32–49.26) | 46.99 | 46.45 | 1.50 (−2.64 to 5.64); 0.489 | 1.53 (−2.63 to 5.68); 0.482 |
| Satisfaction with care at the clinic at 12 mo [6] | | | | | |
| Very satisfied with information given | 141 (46%) | 73 (47%) | 68 (45%) | 4.17 (0.39 to 45.06); 0.239 | 4.45 (0.41 to 48.82); 0.222 |
| Very satisfied with waiting times | 106 (35%) | 59 (38%) | 47 (31%) | 1.67 (0.34 to 8.25); 0.531 | 1.75 (0.35 to 8.6); 0.493 |
| Very satisfied with confidentiality | 155 (50%) | 71 (45%) | 84 (56%) | 194.81 (3.49 to 10887.12); 0.010 | 189.48 (188.7 to 190.27); <0.001 |
| Very satisfied with clinic staff attitude | 142 (46%) | 70 (45%) | 72 (48%) | 12.48 (1.17 to 132.82); 0.036 | 13.28 (1.24 to 142.54); 0.033 |
| Very satisfied with general care | 138 (45%) | 72 (46%) | 66 (44%) | 3.55 (0.32 to 39.17); 0.300 | 3.51 (0.32 to 38.3); 0.303 |

[1] A logistic regression model was fitted for all endpoints, except QoL for which a linear regression was fitted.

[2] These regression analyses include the clustering (clinic) and stratification variables (district) but they are not adjusted for sex.

[3] Approximately 22/307 (8.5%), 11 in control and 11 in intervention, did not have a viral load measurement in the primary endpoint window; these were considered as unsuppressed.

[4] District was removed from regression analysis due to convergence problems.

[5] These regression analyses are in addition adjusted for DTG-based regimen.

[6] These regression analyses are in addition adjusted for the respective baseline measure; 31/157 (20%) in control and 58/150 (39%) in Intervention were missing.

CI, confidence interval; IQR, interquartile range; LTFU, loss to follow-up; mo, months; QoL, quality of life; VL, viral load.

An estimated 2.1 million young people aged 15 to 24 years old are living with HIV in Southern and Eastern Africa [1]. Viral suppression rates among young people living with HIV remain consistently lower than for adults [17] and they are at greater risk of dying and being lost to follow-up than younger children and adults living with HIV [2,3]. Young people have unique needs while navigating through the turbulent phase of adolescence, transitioning to higher schooling or seeking jobs and taking up responsibilities as young adults. Thirty percent of our study participants still attended school and 30% were already married. This calls for DSD, an approach that has received strong policy support from WHO [7,18] and is being piloted in many sub-Saharan African countries [19]. Adolescent-led and -adapted DSD models that offer the entire range of service options (e.g., clinic club, community-based ART refill, SMS notifications, PE support) regularly adapted to the young persons' preferences, and data about their effectiveness remain, however, rare [10].

Three systematic reviews involving studies from 2001 to 2016 evaluated the evidence on interventions to improve engagement in HIV care among adolescents and young adults [6,8,9]. All 3 reviews concluded that despite having worse outcomes than other age groups, only few studies investigating specific interventions for adolescents and young people living with HIV exist and the evidence is of low quality. Nevertheless, 2 reviews cautiously suggest that youth-friendly services and opening hours, multidisciplinary clinics, eHealth support, task-shifting to lay personnel, and peer support and group counselling warrant further research [6,8]. Moreover, a recent meta-analysis specifically assessing psychosocial interventions among adolescents and young adults demonstrated small-to-moderate effects on viral suppression [20].

One of the largest peer-support programs for adolescents living with HIV in sub-Sahara Africa is the Zvandiri program that started in Zimbabwe and engages a cadre of 18- to 24-year-old community adolescent treatment supporters (CATS) [21]. The CATS deliver adherence and psychosocial support at the health facility as well as during a weekly home visit, organize monthly support groups and regular SMS messages, and differentiate the implementation intensity according to HIV vulnerability. This model of care led to improvement in adherence to treatment, retention in care, some psychosocial measures as well as virological outcomes after 2 years [11,22]. The PEBRA trial offered a similar multicomponent DSD model in Lesotho, also in Southern Africa; PEBRA did, however, not show similar effects on viral suppression. Several factors may have contributed to this difference. First, the PEBRA trial ended at 1 year; it is possible that the observed nonsignificant trends towards better outcomes in the intervention group would have increased over a longer time. Second, the Zvandiri trials were conducted exclusively among adolescents living with HIV, i.e., among 10 to 15 years old participants in the first trial and 13 to 19 years old participants in the second trial. Our participants were older, between 15 and 24 years old with a median age of 19 years. Peer-support may be less effective among older participants (20 to 24 years old) who are more likely to be married, have children, and live with their partner. Our pre-planned subgroup analysis suggests such a direction of effect (S4 Table), although this effect has to be interpreted with caution and the interaction term was not significant (p-interaction 0.515). Third, the Zvandiri model of care is more intensive than the PEBRA model of care. For adolescents with an unsuppressed VL or other concomitant health or psychosocial challenges, the CATS conducted 2 to 3 home visits a week, plus weekly phone calls and daily text messages, and a community health nurse or social worker accompanied them. Such differences in a multicomponent intervention may foster different mechanisms to improve outcomes and more research is needed to understand these pathways [23]. Fourth, the Zvandiri trial larger and thus better powered to detect statistical differences.

A literature search revealed 2 other randomized trials that evaluated a support intervention for adolescents living with HIV in sub-Sahara Africa and reported virological outcomes [24,25]. One trial, conducted in Uganda [24], assessed savings-led economic empowerment among 10- to 16-year-old participants, whereas a trial in Zimbabwe [25] proposed a community-based caregiver support program for children and adolescents aged 6 to 15 years with a new HIV diagnosis. Both interventions improved viral suppression rates compared to the standard of care. However, they proposed a different intervention (i.e., economic and caregiver intervention) than the PEBRA model and among a younger population.

Two major events occurred in Lesotho during the trial period: the large-scale rollout of DTG [26] and the first 2 waves of Coronavirus Disease 2019 (COVID-19) [27], which triggered social mobility measures and a brief nationwide lockdown. At enrolment, 12% of the participants were taking a DTG-based regimen. Twelve months later at primary endpoint, this proportion stood at 87%. Importantly, this proportion was equally distributed over both arms.

The COVID-19 pandemic and its local measures made it more difficult for participants to access care and to come for a blood draw assessment, but again, it affected both study groups similarly.

PEBRA trial had several limitations. First, in both groups, 9% of participants in care at 12 months had a missing VL measurement, thus classified as having an unsuppressed VL, which may have underestimated the viral suppression rates. However, the proportion of missing VLs was similar to previous pragmatic trials in the same study districts [28,29]. Second, due to the nature of the study design, the recruiters were aware of the allocation. However, to mitigate recruitment bias among participants, 2 slightly different consent forms for control and intervention were used to conceal the allocation. Third, the data collectors in the control clusters were also young peers, who used the same data collection tool (a restricted version of the PEBRApp). Although they were specifically instructed and were not able to provide the PEBRA model, their presence may have positively influenced follow-up of participants at control clinics. Fourth, the study design and statistical power did not allow for the evaluation of the effectiveness of each individual feature of the PEBRA model, but a descriptive analysis of longitudinal preference data of each component is planned as a follow-up manuscript.

This trial is one of few randomized clinical trials focusing on DSD among young people living with HIV. During the study period of the PEBRA trial, viral suppression rates in both groups increased by 9%. Preference-based peer-coordinated care for young people living with HIV, compared to nurse-based care only, did not lead to conclusive evidence for an effect on viral suppression. More large-scale research is needed to understand the effect of peer-led models of care among the youth.

## Supporting information

**S1 CONSORT Checklist. CONSORT 2010 checklist for cluster-randomized trials.**
(DOCX)

**S1 Table. Details about each PEBRA intervention component option.**
(DOCX)

**S2 Table. Primary analysis model with fixed covariables and prespecified sensitivity analysis with adjustment for baseline VL.**
(DOCX)

**S3 Table. Sensitivity analyses 2 and 3 on primary endpoint.**
(DOCX)

**S4 Table. Primary outcome: Effect modification.**
(DOCX)

## Acknowledgments

We would like to recognize the hard work and valuable contributions of the study staff in all 3 districts, the tireless support of the SolidarMed and Sentebale teams in Lesotho as well as the District Health Management Teams. A special thanks goes to Tlotliso Mafantiri (Tech4All; https://techforall.co.ls/index.html; Lesotho) and Christoph Schwizer (https://schwizer.dev/; Switzerland), who jointly developed the PEBRApp, as well as Ruben Dill for graphic designs. We thank all PEBRA model workshop participants, the PEs, and the involved health facilities for their dedication to this project, and we gratefully acknowledge the young people living with HIV who participated in this trial.

The study is embedded in the SolidarMed and Sentebale country programs and thus benefits from their logistics, administration, and human resources.

## Author Contributions

**Conceptualization:** Mathebe Kopo, Thabo Ishmael Lejone, Nadine Tschumi, Tracy Renée Glass, Mpho Kao, Jennifer Anne Brown, Josephine Muhairwe, Ntoli Moletsane, Niklaus Daniel Labhardt, Alain Amstutz.

**Data curation:** Olivia Seiler, Alain Amstutz.

**Formal analysis:** Nadine Tschumi, Tracy Renée Glass, Olivia Seiler.

**Funding acquisition:** Alain Amstutz.

**Investigation:** Mathebe Kopo, Thabo Ishmael Lejone, Mpho Kao, Jennifer Anne Brown, Olivia Seiler, Alain Amstutz.

**Methodology:** Mathebe Kopo, Thabo Ishmael Lejone, Nadine Tschumi, Jennifer Anne Brown, Josephine Muhairwe, Niklaus Daniel Labhardt, Alain Amstutz.

**Project administration:** Mathebe Kopo, Mpho Kao, Josephine Muhairwe, Ntoli Moletsane, Niklaus Daniel Labhardt, Alain Amstutz.

**Resources:** Mathebe Kopo, Mpho Kao, Ntoli Moletsane, Alain Amstutz.

**Supervision:** Thabo Ishmael Lejone, Ntoli Moletsane, Niklaus Daniel Labhardt, Alain Amstutz.

**Writing – original draft:** Mathebe Kopo.

**Writing – review & editing:** Mathebe Kopo, Thabo Ishmael Lejone, Nadine Tschumi, Tracy Renée Glass, Mpho Kao, Jennifer Anne Brown, Olivia Seiler, Josephine Muhairwe, Ntoli Moletsane, Niklaus Daniel Labhardt, Alain Amstutz.

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
