## [Editor Report · Decision Letter 0]

14 Jul 2022

Dear Dr Amstutz, 

Thank you for submitting your manuscript entitled "Effectiveness of a peer educator-coordinated preference-based differentiated delivery model on viral suppression among young people living with HIV in Lesotho: The PEBRA cluster randomized trial" for consideration by PLOS Medicine.

Your manuscript has now been evaluated by the PLOS Medicine editorial staff and I am writing to let you know that we would like to send your submission out for external peer review.

Please re-submit your manuscript within two working days, i.e. by Jul 18 2022 11:59PM.

Kind regards,

Beryne Odeny

PLOS Medicine

---

## [Decision Letter · Decision Letter 1]

26 Aug 2022

Dear Dr. Amstutz,

Thank you very much for submitting your manuscript "Effectiveness of a peer educator-coordinated preference-based differentiated delivery model on viral suppression among young people living with HIV in Lesotho: The PEBRA cluster randomized trial" (PMEDICINE-D-22-02341R1) for consideration at PLOS Medicine. 

[LINK]

In light of these reviews, I am afraid that we will not be able to accept the manuscript for publication in the journal in its current form, but we would like to consider a revised version that addresses the reviewers' and editors' comments. Obviously we cannot make any decision about publication until we have seen the revised manuscript and your response, and we plan to seek re-review by one or more of the reviewers. 

We expect to receive your revised manuscript by Sep 16 2022 11:59PM. Please email us (plosmedicine@plos.org) if you have any questions or concerns.

We look forward to receiving your revised manuscript. 

Sincerely,

Beryne Odeny, 

PLOS Medicine

plosmedicine.org

1. Author summary - At this stage, we ask that you reformat your non-technical Author Summary. The Author Summary should immediately follow the Abstract in your revised manuscript. This text is subject to editorial change and should be distinct from the scientific abstract. The summary should be accessible to a wide audience that includes both scientists and non-scientists. Please see our author guidelines for more information: https://journals.plos.org/plosmedicine/s/revising-your-manuscript#loc-author-summary.

2. Abstract:

a) Please report your abstract according to CONSORT for abstracts, following the PLOS Medicine abstract structure (Background, Methods and Findings, Conclusions) http://www.consort-statement.org/extensions?ContentWidgetId=562

b) Please rename “Introduction” as “Background.”

c) Background: final section should clearly state what you were comparing across the two delivery models

d) Please provide the number of clinics in each arm (i.e., control and intervention groups)

e) Please quantify the main results (with p values in addition to 95% CI).

f) Please provide the number of participants lost to follow up in each group

g) Please include a summary of adverse events if these were assessed in the study.

h) In the last sentence of the Abstract Methods and Findings section, please describe the main limitation(s) of the study's methodology.

3. Main text: Please rename the Introduction as “Background.”

4. Background: Please describe past research and explain the need for and potential importance of your study. Indicate whether your study is novel and how you determined that. For example, if there has been a systematic review of the evidence related to your study (or you have conducted one), please refer to and reference that review and indicate whether it supports the need for your study.

5. Introduction: Please conclude the introduction with a clear description of the study question or hypothesis including the outcomes of interest.

6. Methods: Please clearly state how many clusters were in each arm

7. Under outcomes, please define what is meant by “loss to follow-up”

8. Please complete the “CONSORT extension for Cluster Trials” and ensure that all components of CONSORT are present in the manuscript and include definition of adherence. When completing the checklist, please use section and paragraph numbers, rather than page numbers.

9. Please add the following statement, or similar, to the Methods: "This study is reported as per the “CONSORT extension for Cluster Trials” guideline (S1 Checklist)."

10. Please clarify whether parental consent was sought for participants aged less than 18 years.

11. Please provide definitions for all abbreviations used in tables and figures and include these as footnotes, e.g., ART, IQR, QoL

12. Results: Please clearly state the number of clusters per arm. 

13. In the main text and tables, please provide both 95% CIs and p values, where appropriate 

14. The terms gender and sex are not interchangeable (as discussed in https://www.who.int/health-

topics/gender); please use the appropriate term.

15. Please remove the “Competing of Interests” statement at the end of the main text. This information is captured in the metadata obtained in the submission form

Comments from the reviewers:

Reviewer #1: Statistical review

This paper reports a cluster randomised trial evaluating a peer-led intervention for improving viral suppression amongst young people with HIV.

Generally the statistical methods and reporting was good. I have some comments below:

1. Abstract: The authors report the primary outcome and a couple of the secondary outcomes. I would recommend adding a statement about other secondaries e.g. 'There were no significant differences between arms in other secondary outcomes', or just report the primary outcome in the abstract. At the moment, it appears selective to report one secondary outcome that had a significant difference.

2. Methods, randomisation: how was the randomisation stratified, using one block in each strata?

3. Statistical analysis: for the sample size calculation, the type I error rate is not given.

4. Statistical analysis, line 225 "(estimated using the delta method)" - I am not sure this is right - the logistic regression gives the confidence interval for odds-ratios.

5. Statistical analysis: could more be added about how missing data was handled for secondary outcomes? 

6. Results - I would recommend that p-values are given alongside CIs in the tables, as the trial was powered for hypothesis testing.

James Wason

Reviewer #2: This paper describes a very well conducted RCT. It is clearly described and methods are sound. 

It approaches an extremely important question - how to improve VL suppression amongst adolescents and young people in Sub-Saharan Africa. 

The trial did (primarily) not produce significant results, but that does not make it any less publishable - in contrast, I think it is important that we do publish these studies, so that we can understand more about what doesn't and does work for this group. 

Minor comments: 

- it would be helpful to give a bit more information about the intervention

- in Table 1, it may be good to check on the terms used 'gay/ lesbian/straight' - it's not my area but plenty of people can give you advice on the best terms to use for this

- from my reading (may be incorrect) both groups improved in terms of VL suppression. I do wonder whether there's a Hawthorne effect here, particularly with the attention given to participants by peer data collectors in the control group. Perhaps just having someone pay attention to them was helpful. I think there is value to saying that the groups both improved (if I read Table 1 correctly)

- A very thoughtful discussion and very interesting that the peer model may work best for adolescents rather than the young people age group. I might suggest adding that pre-specified effect modification analysis into the abstract as it may be important (was it under-powered?) 

Finally, well done to the authors on doing this within a CIPHER grant (which is not a lot of money for an RCT). I would absolutely recommend publishing this. 

LC

Reviewer #3: Many thanks for submitted this manuscript to PLOS. This is an important areas of study given the relatively low rates of VLS amongst adolescents. There are a few issues that I wish to raise for your consideration in revising this manuscript:

1. Why were rural districts/facilities chosen? Would there have been any difference in your results in urban areas were chosen? 

2. Was care provided by public facilities different (especially in the control arm) compared to the missionary led facilities?

3. It is now well known that designing interventions for adolescents younger than 19 (more of whom are presumable still in school and living with care givers) differently to those 20 years and older should be considered - did you consider this? if not why not?

4. Were the activities in the intervention arm co-designed by young people or did the researchers assume that they will make a difference?

Reviewer #4: This is an interesting study on a topic of high importance in the HIV field. Unfortunately, the results did not show a statistically significant impact for the primary outcome. Regardless, I think it is important to report studies like this in high impact journals. That being said, I think there are a number of issues which, if addressed, would substantially improve the paper.

1. Abstract, in the Conclusions, I think "was associated with" rather than "led" may be a more appropriate interpretation.

2. Methods, some more detail on how recruitment was done would be helpful, e.g. was it a convenience sample of whomever showed up at clinic during that study period, etc.?

3. Results, how many young people were screened out and not even approached about possible enrollment in the study?

4. Methods, The role of the app was not exactly clear. Was this app just on the peer's phone? Did participants use the app?

5. Methods, were there no peer educators at all in the control arm?

6. Methods, were study participants compensated?

7. Methods/Results. There was no substantial process data presented which greatly limits interpretation of this study. For example, who actually received the DSD services? What did they choose? How often did they change? How much, if any, did the peers interact with the patients? Without knowing how well the intervention was implemented, it is difficult to draw conclusions.

8. Results, there seems to be a potentially significant difference in age between arms, was adjusting for this considered? The number of children may also be an important difference that might have impacted results.

9. Discussion, is transferring out of care necessarily a bad thing? This seems to be implied throughout, but what evidence is there for this?

10. Results, Table 2 has no p values. There are potentially good reasons for this but I am not sure what the journals standards are for this.

11. Conclusions, it is not clear that the peers were "feasible" because no real feasibility data were presented.

12. Conclusions, not sure I would say this study "supports WHO's recommendation". There are some suggestions of benefit but the primary outcome was negative

13. Conclusions, qualitative findings would be helpful to understand potential benefits not captured by the quantitative results. Was this considered?

[LINK]

---

## [Decision Letter · Decision Letter 2]

7 Nov 2022

Dear Dr. Amstutz,

Thank you very much for re-submitting your manuscript "Effectiveness of a peer educator-coordinated preference-based differentiated delivery model on viral suppression among young people living with HIV in Lesotho: The PEBRA cluster randomized trial" (PMEDICINE-D-22-02341R2) for review by PLOS Medicine.

I have discussed the paper with my colleagues and the academic editor and it was also seen again by xxx reviewers. I am pleased to say that provided the remaining editorial and production issues are dealt with we are planning to accept the paper for publication in the journal.

[LINK]

We look forward to receiving the revised manuscript by Nov 14 2022 11:59PM.   

Sincerely,

Philippa Dodd, MBBS MRCP PhD

PLOS Medicine

plosmedicine.org

Requests from Editors:

Thank you for your considered and detailed responses to previous editor and reviewer comments. There are just a few very minor outstanding editorial requests for you to address as detailed below.

ABSTRACT

Please include p-values where you have reported 95% CIs

Please include the important dependent variables that are adjusted for in the analyses

AUTHOR SUMMARY

Thank you for including an author summary which is very well written and very clearly describes your study. Please could you define IT at first use and SMS (also in the main manuscript text) at first use or use alternative terms if you prefer, I leave it you your discretion.

INTRODUCTION

In the main manuscript text please substitute the heading “Background” with “Introduction”. The term background should only be used in the abstract.

REFERENCES

In the introduction of your main manuscript there are some errors regarding in-text reference callouts. In some instances, a space is lacking between the text and the opening parenthesis for example line 110. In other instances, punctuation precedes the opening parenthesis when it should follow the closing parenthesis, for example line 109. Please check thoroughly throughout the manuscript and amend where necessary.

Comments from the academic editor: 

I agree with the decision for minor revisions. One point to clarify - the authors declare no COIs on the first page of the revised manuscript - I want to be sure not to miss author COIs. 

In light of the above comment please ensure that you have read the journal’s policy regarding competing interests carefully and amend if necessary.

Comments from Reviewers:

Reviewer #1: Thank you to the authors for addressing my previous comments well. I have no further issues to raise.

Reviewer #2: Very good responses. 

Reviewer #4: The authors have done a good job of responding to comments. I have no further comments.

[LINK]

---

## [Editor Report · Decision Letter 3]

14 Nov 2022

Dear Dr. Amstutz,

Thank you very much for re-submitting your manuscript "Effectiveness of a peer educator-coordinated preference-based differentiated delivery model on viral suppression among young people living with HIV in Lesotho: The PEBRA cluster randomized trial" (PMEDICINE-D-22-02341R3) for review by PLOS Medicine.

I am pleased to say that provided the remaining editorial and production issues are dealt with we are planning to accept the paper for publication in the journal.

[LINK]

We look forward to receiving the revised manuscript by Nov 17 2022 11:59PM.   

Sincerely,

Pippa

Philippa Dodd, MBBS MRCP PhD

PLOS Medicine

plosmedicine.org

Requests from Editors:

1) Thank you for including p-values alongside 95% CIs – please define 95% CI at first use in the abstract (line 62) where you have defined adjusted odds ratio.

Thereafter for clarity suggest reporting the statistical information as follows (aOR 4.12; 95%CI [0.45 to 37.62]; p=0.210). Please check and revise accordingly throughout the abstract AND main manuscript.

2) Line 87 (author summary) – “…clinics in Lesotho. In the PEBRA model…” suggest making the sentence beginning “In the PEBRA model…” a separate bulleted point.

3) Table 1: the numerical values continued within parentheses appear to be defined in column headers as “n”. In the table caption these are defined as “(% of those with non-missing data) – which I believe to be the correct definition. Suggest removing the parentheses where participant numbers are detailed in the row headings and placing below the column title (Total, Control, Intervention) to mitigate against confusion here. Please also ensure consistent use of “n” in each row.

4) Table 2: the above applies here also. What do those numerical values within parentheses following “median physical QoL score” depict, for example? All data contained within tables should be easily accessible to the reader. There are many notes in the table to defer the reader to the caption to try and work things out – perhaps too many? Please revise this table accordingly to improve accessibility and clarity. 

In addition, to help improve transparency of data reporting, where you report adjusted analyses in Table 2 please also include the unadjusted analyses. 

5) Supplementary tables: please also include unadjusted analyses for data reported in supplementary files, for example Table S2 and S3. Table S4 - please include include 95% CIs as well as p-values.

6) Line 455: Please remove the heading “conclusion”. The discussion should end with a one paragraph conclusion such that the full discussion reads as one continuous piece of prose, ending in a single paragraph conclusion.

[LINK]

---

## [Editor Report · Decision Letter 4]

17 Nov 2022

Dear Dr. Amstutz,

Thank you very much for re-submitting your manuscript "Effectiveness of a peer educator-coordinated preference-based differentiated delivery model on viral suppression among young people living with HIV in Lesotho: The PEBRA cluster randomized trial" (PMEDICINE-D-22-02341R4) for review by PLOS Medicine.

I have discussed the paper with my colleagues and the academic editor and it was also seen again by xxx reviewers. I am pleased to say that provided the remaining editorial and production issues are dealt with we are planning to accept the paper for publication in the journal.

[LINK]

We look forward to receiving the revised manuscript by Nov 21 2022 11:59PM.   

Sincerely,

Philippa Dodd, MBBS MRCP PhD

PLOS Medicine

plosmedicine.org

Requests from Editors:

We understand and accept your rationale for not including unadjusted analyses for the sensitivity and sub-group analyses presented in the supplementary tables. Thank you for your very clear response which helps to facilitate transparency of data reporting as part of the peer review process. 

We did also request the unadjusted analyses for the primary and secondary outcome measures presented in table 2 but did not see them included. Again, to help us facilitate transparent data reporting, please also include the unadjusted analyses for table 2 ideally in table 2 of the main manuscript (or as a compromise, as a supplementary file). In the table caption please indicate which factors are adjusted for in the analyses.

[LINK]

---

## [Editor Report · Decision Letter 5]

28 Nov 2022

Dear Dr Amstutz, 

On behalf of my colleagues and the Academic Editor, Dr Ruanne Barnabas, I am pleased to inform you that we have agreed to publish your manuscript "Effectiveness of a peer educator-coordinated preference-based differentiated delivery model on viral suppression among young people living with HIV in Lesotho: The PEBRA cluster randomized trial" (PMEDICINE-D-22-02341R5) in PLOS Medicine.

Thank you for your careful attention to previous editorial requests. Alongside the formatting changes (described below), we suggest that table 2 is formatted on the header row to ensure that whole words are not split across multiple lines. It is likely to be possible to achieve this by reducing the width of column one.

PRESS

Best wishes, 

Pippa

Philippa Dodd, MBBS MRCP PhD 

PLOS Medicine